# Signs and Symptoms of Temporomandibular Dysfunction and Radiographic Condylar Morphology in Patients with Idiopathic Condylar Resorption

**DOI:** 10.3390/jcm11154289

**Published:** 2022-07-23

**Authors:** Yanfang Yu, Sijie Wang, Mengjie Wu, Xiaoyan Chen, Fuming He

**Affiliations:** 1Key Laboratory of Oral Biomedical Research of Zhejiang Province, Department of Orthodontics, Stomatology Hospital, School of Stomatology, Zhejiang University School of Medicine, Zhejiang Provincial Clinical Research Center for Oral Diseases, Cancer Center of Zhejiang University, Hangzhou 310027, China; ojyu@163.com (Y.Y.); 22018590@zju.edu.cn (S.W.); wumengjie@zju.edu.cn (M.W.); 2Key Laboratory of Oral Biomedical Research of Zhejiang Province, Department of Prosthodontics, Stomatology Hospital, School of Stomatology, Zhejiang University School of Medicine, Zhejiang Provincial Clinical Research Center for Oral Diseases, Cancer Center of Zhejiang University, Hangzhou 310027, China

**Keywords:** idiopathic condylar resorption, signs and symptoms, cone beam computed tomography, condylar morphology

## Abstract

Background: Little is known about the clinical characteristics of idiopathic condylar resorption (ICR). The aim of this study was to examine the signs and symptoms of temporomandibular dysfunction (TMD) and evaluate the morphological characteristics of the condyles in patients with ICR. Methods: Sixty patients with ICR (41 in the bilateral ICR group and 19 in the unilateral ICR group) and forty-one healthy controls were examined. Signs and symptoms of TMD were described, and three-dimensional models of the condyles were measured and analyzed. Results: In total, 81.7% of ICR patients had self-reported symptoms and 78.3% of ICR patients had objective-found signs. The anteroposterior diameter, transverse diameter, height, maximal sectional area, volume of the condyles, axial angle, and the distance from the posterior point of the condyle to the Saggittal standard line were significantly smaller in the ICR condyles compared with the controls (*p* < 0.05). The condylar neck angle was significantly larger in the ICR condyles compared with the controls (*p* < 0.05). Conclusions: Most patients with ICR had signs and symptoms of TMD. The prevalence of clicking and opening–closing deviation was significantly different between the bilateral and the unilateral ICR groups. In patients with ICR, the size of the condyles decreased significantly; the condyles also rotated inward, moved forward, and inclined posteriorly.

## 1. Introduction

Idiopathic condylar resorption (ICR) manifests as progressive condylar resorption with an unknown cause, which results in pain, malocclusion, and esthetic changes. Previous studies have shown that ICR predominantly affects women in their second to third decades of life [1,2], especially during the pubertal growth spurt [3,4]. ICR is usually bilateral but can also be unilateral. In bilateral ICR, the jaw exhibits a decreased vertical ramus height, a retruded mandible, and an increased mandibular plane angle and anterior face height, resulting in an anterior open bite or class II malocclusion [5]. In patients with unilateral ICR, the vertical heights of the mandibular body and ramus and the occlusal plane on each side are different; these patients present with mandibular deviation and facial asymmetry [6].

Many possible pathogenic factors of ICR have been proposed previously. A study by Arnett et al. in 1996 reported that the mechanism of ICR was condylar dysfunctional remodeling [5,7]. They speculated that the primary causes of ICR were reduced self-remodeling capacity and increased mechanical stress. Sex hormones, such as 17 β-estradiol (E2), are one of the factors of reduced self-healing ability of the condyles [7]. Mechanical stress of the condyle could increase due to factors such as internal joint derangement, trauma, parafunctional habits, orthodontics, orthognathic surgery, and other occlusal therapies [8].

It is difficult to diagnose and treat ICR. Previous studies have reported that the most common symptoms of ICR are joint clicking, pain, and restricted mouth opening. One-quarter (25%) of the patients with ICR exhibited no symptoms of temporomandibular dysfunction (TMD) [6], and little is known about the clinical characteristics of ICR. Therefore, further research about the signs and symptoms of ICR is necessary.

CBCT and MRI are valuable for TMJ evaluation. MRI remains as the gold standard examination for TMJ features analysis due to its high-contrast sensitivity to tissue differences and the absence of ionizing radiation. MRI was most specific and sensitive for the interpretation of soft tissue and inflammatory conditions in the joint, whereas CBCT examination produced excellent images for osseous morphology and pathology [9].The joint space, sclerosis, flattening margins, and bony collapse at articular surfaces are seen clearly on CBCT images [10]. CBCT can also be used to analyze a 3D configuration of the TMJ [10,11,12]; the condylar head volume and the extent of bone loss can be quantified with CBCT [10,12]. CBCT images obtained in series can be used to assess whether ICR is active or arrested. Loss of condyle volume means ICR was active between two capture dates. If no volume was further lost, one can conclude that ICR is no longer active [12]. This is particularly important for the treatment of ICR.

The aim of this study was to examine the clinical features of ICR such as susceptible population, the signs and symptoms of TMD and the detailed changes of condylar morphology in patients with ICR. We hypothesized that ICR has characteristic clinical and imaging manifestations.

## 2. Materials and Methods

### 2.1. Participants

This retrospective observational study was approved by the Medical Ethics Committee of Stomatology Hospital, Zhejiang University School of Medicine (Approval No. 07). The archives of the patients from the Department of Orthodontics and TMJ specialist clinic of Stomatology Hospital, Zhejiang University were screened by an experienced doctor from 2014 to 2020, and all eligible patients were included in the study. The patients were diagnosed with ICR according to the recommended diagnostic criteria proposed by Peck et al. [13] The patients reported to our hospital with malocclusion, maxillofacial aesthetic changes, and TMD, and they had short-term progressive occlusal and facial aesthetic changes. Cone-beam computed tomography (CBCT) imaging showed significant condylar bone destruction with an unknown cause. The exclusion criteria for the ICR group were (1) rheumatoid diseases, (2) developmental malformation, (3) history of craniofacial trauma or surgery, and (4) previous orthodontic treatment. The control group included patients scheduled for general orthodontic treatment in the Department of Orthodontics. Only patients whose CBCT records showed normal condylar images were included. CBCT imaging was used to evaluate the root bone relationship and perform three-dimensional (3D) cephalometric analysis. The exclusion criteria for the control group were (1) signs and symptoms of TMD, (2) rheumatoid diseases, (3) developmental malformation, (4) history of craniofacial trauma or surgery, and (5) previous orthodontic treatment. The participants included 60 Chinese patients with ICR (the ICR group: mean age, 20.9 ± 4.3 years; 57 females, 3 males) and 41 age- and gender-matched healthy controls (the control group: mean age, 20.4 ± 4.5 years; 39 females, 2 males). The ICR group was divided into two subgroups: 41 bilateral ICR cases were assigned to the bilateral ICR group (mean age, 20.9 ± 4.4 years; 39 females, 2 males) and 19 unilateral ICR cases were assigned to the unilateral ICR group (mean age, 21.1 ± 4.2 years; 18 females, 1 male).

### 2.2. Signs and Symptoms

Signs and symptoms of ICR were obtained through clinical consultation and examination [14] performed by an experienced doctor within one month after CBCT examination. Patients without any signs and symptoms of TMD were recruited into the control group. The symptoms of TMD include (1) joint sounds: clicking and crepitation, (2) orofacial pain: temporomandibular joint (TMJ) and masticatory muscle pain, and (3) TMJ function: joint locking and limited mouth opening. The signs of TMD include (1) joint sounds: clicking and crepitation; (2) orofacial pain: TMJ and masticatory muscle pain on palpation, and (3) TMJ function: maximal opening and deviation during jaw opening and closing. Maximum interincisal opening less than 35 mm was defined as limited mouth opening.

### 2.3. CBCT Images

All CBCT images were obtained using the New Tom VGI device (Quantitative Radiology, Verona, Italy) with the following parameters: 110 kVp, 2.0 mA, 15 × 15 cm field of view, 0.3 mm voxel size, and 40 s scanning time. The participants stood and bit their teeth into the maximum intercuspal position. Their heads were positioned with the Frankfort horizontal (FH) plane parallel to the floor.

### 2.4. Image Measurements

All image processing and measurements were completed by one examiner who can use the Dophin software skillfully.

Raw CBCT data were exported in the Digital Imaging and Communications in Medicine (DICOM) format and reconstructed by the Dolphin Imaging program (version 11, Dolphin Imaging and Management Solutions, Chatsworth, CA, USA). Image orientation was conducted by rotating the 3D reconstructed image to align with the FH plane horizontally and both midsagittal and transporionic planes vertically (Figure 1). Condylar characteristics were evaluated using axial and sagittal reconstructions. The axial slice of the maximum mediolateral view of the condyle was selected for measurement (the interval between the slices was 0.5 mm). The measurement parameters included anteroposterior diameter, transverse diameter, axial angle, axial distance (the distance from the center of the condylar axis (the intersection of LCo-MCo and ACo-PCo) to the midsagittal line), and maximal sectional area (Smax) (Figure 2).

Two-dimensional radiographic images were created from the 3D data using the “Build X-ray” function. Lateral cephalometric radiographs were constructed from the left and right sides of the 3D images (Figure 3). The measurement parameters of the sagittal plane included the height of the condyle, the angle of the condylar neck, and the distance from the posterior point of the condyle (Pcd) to the line perpendicular to the FH plane, passing through the sella point (Pcd-FH-*p*(S)) (Figure 4). The measurements of condylar neck angle were performed according to the study by Soon-Jung H et al. [15].

To eliminate the interference caused by the inclination of the ramus, the 3D skull image was reoriented before condylar volume measurement. The 3D image was rotated to align the mandibular ramus vertically. The inferior limit of the condyle was determined by passing the horizontal plane through the deepest point of the sigmoid notch. The condyles were delimited and cropped. Semi-automated segmentation of the 3D model was established. The reorientation and condylar cropping methods were performed according to the study by Silva et al. [16]. The volume of each condyle was displayed in cubic millimeters (mm^3^) (Figure 5).

Definition of landmarks and measurement items are shown in Table 1.

### 2.5. Statistical Analysis

To assess the reliability of the measurements, 25 patients with 50 TMJs were randomly selected for repeated measurements. To ensure intraobserver reliability, the measurements were repeated by the same examiner after 2 weeks, and for interobserver reliability, another investigator performed the same measurements after 2 weeks. The intraobserver (correlation coefficients range, 0.926–0.987) and interobserver reliabilities (correlation coefficients range, 0.924–0.985) were excellent (all, >0.90). Statistical analyses were performed using SPSS software, version 26 (IBM Co., Armonk, NY, USA). Descriptive statistics are presented as mean ± standard deviation. The distribution of each data set was analyzed using the Shapiro–Wilk test. An independent T-test was used to analyze normally distributed data, while the Mann–Whitney U-test was used to analyze non-normally distributed data. The Chi-square and Fisher exact tests were used to determine the statistical differences in signs and symptoms prevalence between the two groups, and *p* < 0.05 was considered statistically significant.

## 3. Results

### 3.1. Bilateral ICR Group

The proportion of patients with ICR who self-reported their symptoms or had a history of self-reporting their symptoms were 82.9% in the bilateral ICR group. Clicking, TMJ pain, joint locking, crepitation, and limited mouth opening were the common TMD symptoms. Clicking was found in 78.0% of the patients and 69.5% of the joints in the bilateral ICR group. TMJ pain was found in 36.6% of the patients and 30.5% of the joints in the bilateral ICR group. Joint locking was found in 19.5% of the patients and 15.9% of the joints in the bilateral ICR group (Table 2). Our data showed that 73.2% of the patients with ICR had objective-found signs in the bilateral ICR group. The common signs were opening–closing deviation, crepitation, clicking, maximum opening limitation, and TMJ tenderness. In the bilateral ICR group, 48.8% of the patients had opening–closing deviation. Crepitation was identified in 56.1% of the patients and 46.3% of the joints in the bilateral ICR group. Most of the measurements variables of the condyle were normally distributed (Shapiro–Wilk test, *p* > 0.05) except for the data on axial angle in the bilateral ICR group and on condylar neck angle in the control group (*p* < 0.05) (Table 3).

The anteroposterior diameter, transverse diameter, height, maximal sectional area, condyle volume, axial angle, and Pcd-FH-*p*(S) were significantly smaller, and the condylar neck angle was significantly larger in the bilateral ICR group compared with the control group (*p* < 0.05) (Table 4).

### 3.2. Unilateral ICR Group

The proportion of patients with ICR who self-reported their symptoms or had a history of self-reporting their symptoms was 78.9% in the unilateral ICR group. Clicking was found in 73.7% of the patients and 50% of the joints in the unilateral ICR group. The prevalence of clicking in the joints was significantly higher in the bilateral ICR group than in the unilateral ICR group (*p* < 0.05). TMJ pain was found in 36.8% of the patients and 26.3% of the joints in the unilateral ICR group. Joint locking was found in 31.6% of the patients and 23.7% of the joints in the unilateral ICR group. However, there was no statistically significant difference in the overall prevalence of these two symptoms between the bilateral and unilateral ICR groups (*p* > 0.05) (Table 2). Our data showed that 89.5% of the patients with ICR had objective-found signs in the unilateral ICR group. In the unilateral ICR group, 78.9% of the patients had opening–closing deviation. The prevalence of opening–closing deviation was significantly higher in the unilateral ICR group than in the bilateral ICR group (*p* < 0.05). Crepitation was identified in 47.4% of the patients and 34.2% of the joints in the unilateral ICR group. This result was not significantly different between the groups (*p* > 0.05) (Table 3).

In the unilateral ICR group, the anteroposterior diameter, transverse diameter, condylar height, maximal sectional area, condyle volume, axial angle, and Pcd-FH-*p*(S) were smaller, and the condylar neck angle was larger on the side with ICR compared with the control group; the difference was significant (*p* < 0.05). When the healthy side of patients in the unilateral ICR group was compared with the control group, the anteroposterior diameter, transverse diameter, and maximal sectional area were significantly smaller in the unilateral ICR group (*p* < 0.05) (Table 4).

## 4. Discussion

Previous studies have found that ICR was common in females aged 10–40 years [6]. Condylar resorption has always been associated with cosmetic problems, and a large portion of patients with ICR visited the hospital to improve their appearance but not to improve the function of the condyle. Considering the extended time it takes for esthetic changes caused by condylar resorption to begin manifesting, the real age at TMD onset might probably be earlier than the age at which it is mostly calculated. In our study, 31 out of 60 patients with ICR could provide information on the accurate age at which TMD symptoms occurred, and the average age was 16.8 ± 3.4 years (range, 12 to 26 years). This information is important for the clinical characteristics of ICR.

There have been few previous studies on the signs and symptoms of TMD in patients with ICR. Kristensen et al. [17] found that 20% of patients with ICR had no TMJ arthralgia, myalgia, or TMJ sounds. Wolford et al. [6] reported that 25% of patients with ICR had no signs or symptoms of TMD. In this study, we found that 81.7% and 78.3% of the patients with ICR had self-reported symptoms and objective-found signs, respectively. These results are consistent with previous studies. Most of the patients with ICR had signs and symptoms of TMD.

There are many etiologies of joint sounds such as disc displacement, cavitation, reduced synovial fluid within the joint, infrequent joint mobilization, and others. However, the most common cause of joint clicking is disc displacement. Kristensen et al. [14] studied the signs and symptoms of 25 patients with ICR and found that 52% of the patients and 42% of the joints had the clicking symptom. Previous studies have reported that the prevalence of TMJ sounds in patients with progressive condylar resorption (PCR) was 38% to 76% [15,18,19]. Our results showed that clicking prevalence in joints in the bilateral ICR group was higher than that in the unilateral ICR group (*p* < 0.05). We found that some patients with bilateral ICR had unilateral joint clicking, and some patients with unilateral ICR had bilateral joint clicking. Interestingly, 76.7% of the patients with ICR reported a history of clicking, but only 15.0% of the patients reported that the sign of clicking was identified by examination. This may be due to condyle/fossa remodeling during the development of ICR, which may have improved the disc–condyle relationship.

In our study, about 30% of the patients with ICR suffered from TMJ pain. This result is similar to those of previous studies. Kristensen et al. [15] reported that TMJ pain was found in 40% of the patients with ICR. The findings by Handelman and Greene [20] showed that one-third of the patients with ICR (30.8%) had TMJ pain. The pain intensity is usually approximately 3–4 out of 10 [6,8]. The main reason for TMJ pain was probably the irritation to the nerves caused by abnormal TMJ structure and the dysfunction of relevant muscles. Patients with ICR were very sensitive to pain, although pain was not the most common symptom.

TMJ locking was reported in 28% of patients with ICR in the study by Kristensen et al. [17]. Kerstens et al. [18] reported that 22% of the patients with PCR had the TMJ locking symptom. Our results showed that 19.5% of the patients and 15.9% of the joints in the bilateral ICR group had joint locking, while 31.6% of the patients and 23.7% of the joints had joint locking in the unilateral ICR group. However, the difference between the two groups was not statistically significant (*p* > 0.05). We also found that 16.7% of the patients with ICR self-reported that they had crepitation and 11.7% of the patients had reduced mouth opening, which are consistent with the results of Kristensen et al. [17], reporting that 16% of the patients with ICR had reduced mouth opening.

In our study, the prevalence of opening–closing deviation in the unilateral ICR group was obviously higher than that in the bilateral group (*p* < 0.05). The reason for opening–closing deviation could be the difference in joint mobility, especially in the patients with unilateral ICR, and the proportion of objective-found crepitation was 53.3% in the patients with ICR, which was extremely higher than the 28% reported in Kristensen et al.’s study [17]. Crepitation indicates bony destruction [21].

The transverse diameter, anteroposterior diameter, and the condylar height were obviously smaller in the ICR condyles compared to the healthy condyles, which suggests that resorption occurred in the vertical, sagittal, and horizontal directions. The most significant reduction was a reduction in condylar height. We found that the average height of the ICR condyles in the bilateral and unilateral ICR groups was 5.33 mm and 5.88 mm lower than that of condyles in the control group. Meanwhile, we found that the average Smax in the bilateral ICR group and unilateral ICR group was 38.95 mm^2^ and 44.60 mm^2^ smaller than that in the controls, and the average condylar volume in the bilateral ICR group and unilateral ICR group was 649.25 mm^3^ and 733.00 mm^3^ smaller than that in the controls, respectively. Yi fan et al. [22] studied the condylar Smax and volume and reported similar findings. However, the results were not comparable due to different measuring methods. Clinically, excessive mechanical loading on the condylar tissues leads to the condylar resorption, and subsequently, the decreased condylar area and volume might lead to excessive mechanical loading on the condylar tissues, which would cause further condylar resorption [17,23].

Interestingly, we found that the anteroposterior diameter, transverse diameter, and maximal sectional area were significantly smaller on the healthy side of the condyle of the patients in the unilateral ICR group compared with the control group (*p* < 0.05). We speculated that systemic pathogenic factors of ICR can influence the development and remodeling of condyles on the healthy side. Another possible reason was that ICR was more likely to occur in patients with smaller condyles.

Many morphological variations of the condyle were found in the patients with ICR in this study. Park et al. [24] reported a mean decrease of 5.7° in the condylar axial angle in patients undergoing orthognathic surgery for condylar remodeling. Kristensen et al. [17] reported a mean decrease of 10.6 degrees in the condylar axial angle of patients with ICR. Anatomically, the condylar head emerges from the ramus and rotates outward in a superior direction, so the change in axial angle may be an index to evaluate the severity of condylar resorption. Few previous studies have been published on the changes in condylar neck angle in patients with ICR. Yi fan et al. [22] measured condylar neck angles and reported similar findings. The results of this study suggested that the condyle rotates inward and inclined posteriorly after resorption. Such morphological variations of the condyle will probably change the direction of stress on the condyles and subsequently influence its growth and remodeling.

The results of the Pcd-FH-*p*(S) were significantly smaller in the ICR condyles compared with the condyles of the controls. There was no significant difference in axial distance between the ICR group and the control group. Previous studies [25,26] have reported that the mandible exhibits decreased vertical ramus height and posterior facial height and backward rotation in patients with ICR. In the development of ICR, the morphological and size changes of the mandible and condyle as well as the breaking and reconstruction of muscle balance moves the condyle to a relatively anterior position. The mechanism is extremely complicated and needs further research.

As indicated above, ICR is a well-known but poorly understood syndrome with multifactorial etiology that predominantly affect young women. The diagnosis and treatment of ICR is challenging. The results of this study provide a deeper understanding of the clinical characteristics of ICR. The characteristics of the signs and symptoms of TMD, the real age at TMD symptom onset, the most common signs and symptoms, the male-to-female ratio of ICR, and others are helpful for the early diagnosis and risk assessment of high-risk populations. The results of 3D morphological study of the condyles are helpful for imaging diagnosis and determination of absorption severity.

We comprehensively evaluated the morphological and positional characteristics of the condyles in patients with ICR in this study. We are the first to study the different manifestations of symptoms and signs of bilateral ICR group and unilateral ICR group separately. However, there were some limitations in this study. (1) There was possible selection bias: patients attending the hospital might have had a higher prevalence of the symptoms of TMD than that observed in patients with ICR in a population. (2) Signs and symptoms of ICR were obtained from clinical consultation and examination, but condylar resorption may have been static when some of the patients visited the hospital, which may have led to obtaining incomplete information from the consultation. (3) The sample size is not large enough for epidemiological analysis. (4) Most of the measurements in this study were 2D data; condylar morphology should be deeply studied with 3D data. More clinical and scientific research about ICR should be performed in the future to better understand ICR and identify appropriate diagnosis and optimal treatment.

## 5. Conclusions

Most of the patients with ICR had signs and symptoms of TMD. Clicking, TMJ pain and joint locking were the common TMD symptoms. Opening–closing deviation, crepitation, and clicking were the common TMD signs. The prevalence of clicking and opening–closing deviation were significantly different between the bilateral and the unilateral ICR groups. The size of ICR condyles decreased significantly in patients with ICR compared with controls. The most significant reduction was the condylar height. The condyles rotate inward, moved forward, and inclined posteriorly in patients with ICR.

## Figures and Tables

**Figure 1 jcm-11-04289-f001:**
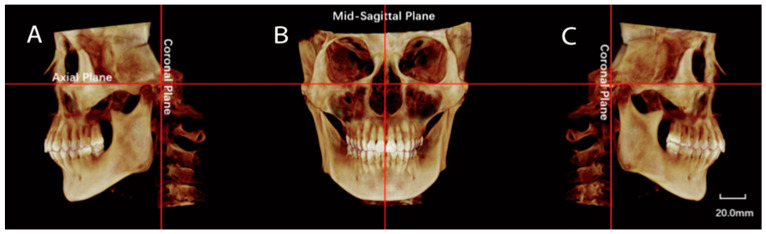
Orientation of the CBCT scan before measurements. (**A**,**C**): Alignment of the Frankfort-horizontal (FH) plane and transporionic vertical plane from the left (**A**) and right (**C**) view of 3D reconstruction. (**B**): Alignment of the FH plane and the midsagittal plane from the front view.

**Figure 2 jcm-11-04289-f002:**
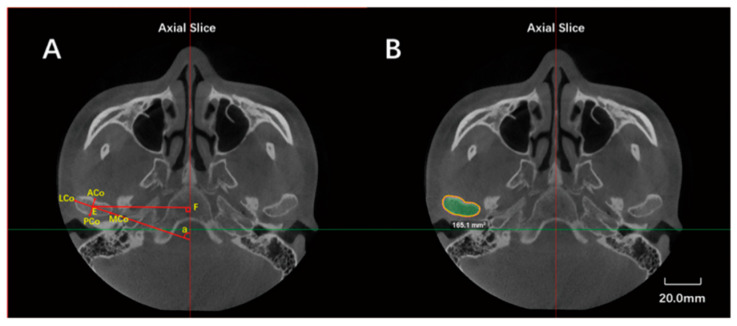
Measurements in the axial view of the condyle. (**A**): Anteroposterior diameter—the distance from ACo to PCo; transverse diameter—the distance from LCo to MCo; the axial angle (∠α: the angle between the line connecting LCo to MCo and the midsagittal line); the axial distance (EF)—the perpendicular distance from the condylar axial center point E (the intersection of LCo-MCo and ACo-PCo) to midsagittal line. (**B**): Maximal sectional area (Smax)—the axial condylar area in this axial slice. (ACo: Most anterior point of the condyle; PCo: Most posterior point of the condyle on axial plane; LCo: Most lateral point of the condyle; MCo: Most medial point of the condyle).

**Figure 3 jcm-11-04289-f003:**
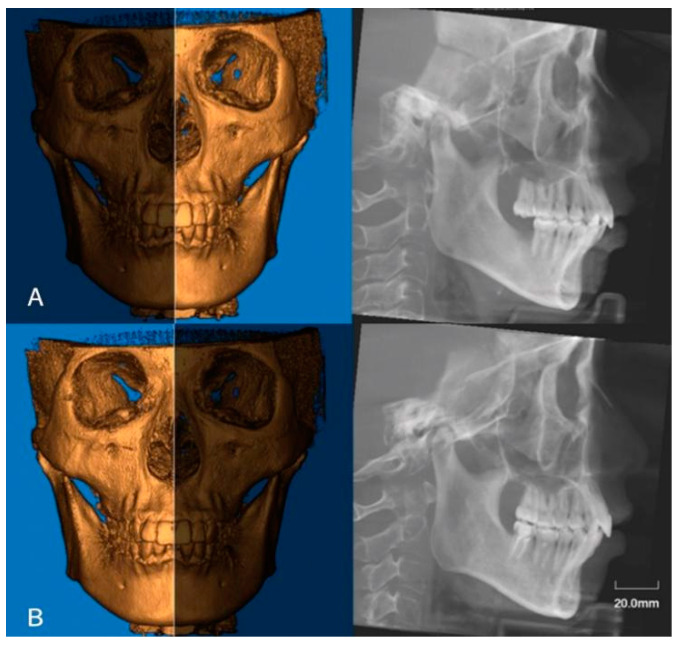
Lateral cephalometric radiographs were constructed from left side (**A**) and right side (**B**), respectively.

**Figure 4 jcm-11-04289-f004:**
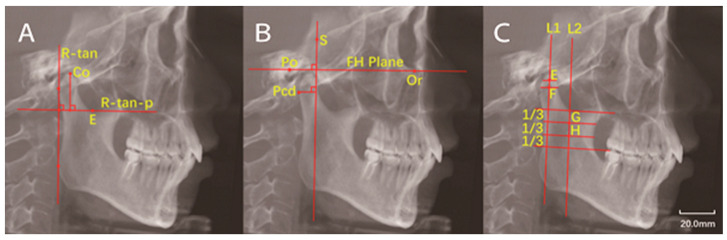
Measurements in the sagittal view of the condyle. (**A**): Condylar height—the distance from Co to the line perpendicular to the R-tan line and passing the deepest point of the sigmoid notch (R-tan-*p*). (**B**): Pcd-FH-*p*(S)—the distance from Pcd to the line perpendicular to FH plane and passing S point. (**C**): Condylar neck angle—the center of the widest width of the condyle (F point) and the center of the middle part between the top and the widest width of the condyle were marked (E point); a line was drawn through these two points. The midpoints of the width of the superior border (G point) and the inferior border (H point) of the middle third of the ascending ramus were determined. Another line was drawn through these two midpoints. The angle between these two lines was defined as condylar neck angle. The angle was defined as positive if two lines intersected below the condyle and negative if they intersected above the condyle. (Co: Most superior point of the condyle; R-tan line: Tangent to the posterior border of the ramus; Pcd: Most posterior point of the condyle on sagittal plane; FH plane: Frankfort-horizontal plane; S point: Sella point; Po: The uppermost point of external auditory canal; Or: The lowest point of orbital margin.)

**Figure 5 jcm-11-04289-f005:**
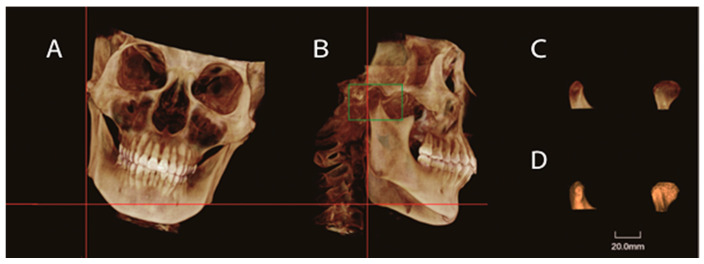
Condylar volume measurements. (**A**,**B**) Reorientation of the CBCT image before the measurements of condylar volume. The 3D image was rotated to align the mandibular ramus vertically. The inferior limit of the condyle was determined by the horizontal plane passing the deepest point of the sigmoid notch. (**C**,**D**): Condyles were delimited and cropped. The volume of each condyle was displayed in cubic millimeters (mm^3^).

**Table 1 jcm-11-04289-t001:** Definition of landmarks and measurement items.

Landmarks, Constructed Lines and Measurement Items	Abbreviation	Definition
Lateral condylar point	LCo	Most lateral point of the condyle
Medial condylar point	MCo	Most medial point of the condyle
Anterior condylar point	ACo	Most anterior point of the condyle
Superior condylar point	Co	Most superior point of the condyle
Posterior condylar point	PCo	Most posterior point of the condyle on axial plane
Posterior condylar border point	Pcd	Most posterior point of the condyle on sagittal plane
Sella	S	The center of sella on the median sagittal plane of skull
Porion	Po	The uppermost point of external auditory canal
Orbitale	Or	The lowest point of orbital margin
Ramus tangent line	R-tan	Tangent to the posterior border of the ramus
Ramus tangent line perpendicular	R-tan-*p*	Line perpendicular to R-tan and tanging the deepest point of the sigmoid notch
Frankfort horizontal plane	FH plane	Line from Po to Or
Anteroposterior diameter	ACo-PCo	Distance from ACo to PCo
Transverse diameter	LCo-MCo	Distance from LCO to MCo
Condylar height	-	Vertical distance from Co to R-tan-*p*
Maximal sectional area	Smax	Maximal sectional area of the condyle
Axial angle	-	Angle between LCo-MCo and midsagittal line
**-**	FH-*p*(S)	Line perpendicular to FH plane and through the sellar point
**-**	Pcd-FH-*p*(S)	Vertical distance from Pcd to FH-*p*(S)

**Table 2 jcm-11-04289-t002:** Frequencies and percentages of self-reported symptoms in the ICR group.

	Joints				Patients			
	ICR Group	Bilateral ICR Group	Unilateral ICR Group	*p*	ICR Group	Bilateral ICR Group	Unilateral ICR Group	*p*
Clicking	76/120 (63.3%)	57/82 (69.5%)	19/38 (50.0%)	0.039 *	46/60 (76.7%)	32/41 (78.0%)	14/19 (73.7%)	0.965
TMJ pain	35/120 (29.2%)	25/82 (30.5%)	10/38 (26.3%)	0.640	22/60 (36.7%)	15/41 (36.6%)	7/19 (36.8%)	0.985
Joint locking	22/120 (18.3%)	13/82 (15.9%)	9/38 (23.7%)	0.302	14/60 (23.3%)	8/41 (19.5%)	6/19 (31.6%)	0.484
Crepitation	16/120 (13.3%)	13/82 (15.9%)	3/38 (7.9%)	0.233	10/60 (16.7%)	8/41 (19.5%)	2/19 (10.5%)	0.620
Limited mouth-opening	-		-		7/60 (11.7%)	5/41 (12.2%)	2/19 (10.5%)	1.000
Symptoms	-		-		49/60 (81.7%)	34/41 (82.9%)	15/19 (78.9%)	0.990

ICR: Idiopathic condylar resorption; TMJ: Temporomandibular joint. * Bilateral ICR group vs. unilateral ICR group *p* < 0.05.

**Table 3 jcm-11-04289-t003:** Frequencies and percentages of objective-found signs in ICR group.

	Joints				Patients			
	ICR Group	Bilateral ICR Group	Unilateral ICR Group	Sig	ICR Group	Bilateral ICR Group	Unilateral ICR Group	Sig
Opening–closing deviation	-	-	-		35/60 (58.3%)	20/41 (48.8%)	15/19 (78.9%)	0.027 *
Crepitation	51/120 (42.5%)	38/82 (46.3%)	13/38 (34.2%)	0.211	32/60 (53.3%)	23/41 (56.1%)	9/19 (47.4%)	0.528
Clicking	13/120 (10.8%)	11/82 (13.4%)	2/38 (5.3%)	0.307	9/60 (15.0%)	7/41 (17.1%)	2/19 (10.5%)	0.786
Maximum opening limitation	-	-	-		3/60 (5.0%)	2/41 (4.9%)	1/19 (5.3%)	1.000
TMJ and muscle tenderness	1/120 (0.8%)	0 (0.0%)	1/38 (2.6%)	0.317	1/60 (1.7%)	0 (0.0%)	1/19 (5.3%)	0.317
Signs	-	-	-		47/60 (78.3%)	30/41 (73.2%)	17/19 (89.5%)	0.276

ICR: Idiopathic condylar resorption; TMJ: Temporomandibular joint. * Bilateral ICR group vs. unilateral ICR group *p* < 0.05.

**Table 4 jcm-11-04289-t004:** Measurements of condylar size, morphology and position in ICR group and control group.

	Control Group (*n* = 41)	Bilateral ICR Group (*n* = 41)	Resorption Side in Unilateral ICR Group (*n* = 19)	Healthy Side in Unilateral ICR Group (*n* = 19)
	Mean ± SD	Mean ± SD	Mean ± SD	Mean ± SD
Anteroposterior diameter (mm)	8.07 ± 0.87	6.43 ± 1.11 ***	6.27 ± 1.06 ***	7.51 ± 1.13 *
Transverse diameter (mm)	18.70 ± 1.97	15.44 ± 2.99 ***	14.81 ± 1.84 ***	16.93 ± 1.90 **
Condylar height (mm)	21.22 ± 3.19	15.89 ± 2.86 ***	15.34 ± 1.39 ***	21.06 ± 3.47
Smax (mm^2^)	127.67 ± 18.27	88.72 ± 19.03 ***	83.07 ± 13.66 ***	107.77 ± 21.19 ***
Condylar volume (mm^3^)	1626.74 ± 406.71	977.49 ± 302.50 ***	893.74 ± 209.87 ***	1442.79 ± 459.13
Axial angle (°)	71.85 ± 6.18	54.60 ± 10.53 ***	55.58 ± 5.33 ***	68.97 ± 6.83
Condylar neck angle (°)	0.99 ± 7.38	20.18 ± 7.13 ***	18.34 ± 8.53 ***	4.25 ± 7.29
Axial distance (mm)	51.30 ± 2.40	51.30 ± 2.40	52.51 ± 1.94	53.71 ± 2.19
Pcd-FH-*p*(S) (mm)	17.11 ± 2.41	13.87 ± 3.14 ***	12.67 ± 2.54 ***	15.84 ± 3.33

vs. control group *** *p* < 0.001; ** *p* < 0.01; * *p* < 0.05.

## Data Availability

Data available on request due to restrictions, e.g., privacy or ethical.

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
