# Peer review of "Signs and Symptoms of Temporomandibular Dysfunction and Radiographic Condylar Morphology in Patients with Idiopathic Condylar Resorption"

_jcm, 2022, doi:10.3390/jcm11154289_

Round 1
Reviewer 1 Report
Dear authors,
I am glad that there is a growing interest for researchers about idiopathic condylar resorption. The study should be read by a native English speaker. I would suggest to modify the title, as the study is more focused on "radiographic condylar morphology" and self-reported signs and symptoms.
Few suggestions:
Title: it should contain the type of the study.
Abstract:
"Sagittal" instead of "sagittal".Page 1, lines 35-41: the sentence should be shortened.The conclusions should be organized for unilateral and bilateral ICRs.
Introduction: I would suggest you to define ICR according to the TMD-DC.
Page 2, lines 52-53: "it's difficult to define" should be omitted.
Aim of the study: please specify the "clinical features" in the aim of the study.
Methods:
Page 2, line 81: the name of the ethical committee is not reported.
The sample size has been calculated?
The condyle measurement protocol/cephalometric analysis have been proposed/validated by previous studies?
Results:
Two subheadings should be created: unilateral and bilateral ICR. It should be interesting to correlate the radiographic findings (consular height) with clinical signs (open bite, mandibular deviations). You assessed three-dimensionally just the volume; the rest of the analysis consisted of 2D measurements. I believe that the condylar morphology should be deeply studies with 3D data.
Discussion:
It should be divided in three sections: interpretation of the results, generalization of the results in the context of the available literature, strengths and limitations of the study. The importance of CBCT is clear and evident.
CONCLUSION: I do not understand how you demonstrated that the condyles were twisted in ICR
The study is not reported according to the Strobe guidelines
Author Response
Reviewer 1
|
|
I am glad that there is a growing interest for researchers about idiopathic condylar resorption. The study should be read by a native English speaker. I would suggest to modify the title, as the study is more focused on "radiographic condylar morphology" and self-reported signs and symptoms.
Title: it should contain the type of the study.
Response 1: Thank you for the revise opinion. I do agree that this study is more focused on"radiographic condylar morphology", therefore, I revised the title to “Signs and symptoms of temporomandibular dysfunction and radiographic condylar morphology in patients with idiopathic condylar resorption.” About the suggestion “title shoud be contain the type of the study”, In my opinion this study is a descriptive study and the title should be long if added the type.
Abstract:
"Sagittal" instead of "sagittal".
Response 2:The first letter of “sagittal”was capitalized.
Page 1, lines 35-41: the sentence should be shortened.
Response 3:Line 35-41: the sentence has been shortened to”81.7% ICR patients had self-reported symptoms and 78.3% ICR patients had objective-found signs. The anteroposterior diameter, transverse diameter, height, maximal sectional area, volume of the condyles, axial angle, and the distance from the posterior point of the condyle to the Saggittal standard line were significantly smaller in the ICR condyles compared with the controls (P<0.05). The condylar neck angle was significantly larger in the ICR condyles compared with the controls (P<0.05).”
The conclusions shoulde be organized for unilateral and bilateral ICRs.
Response 4:There was no significant difference about morphological changes in condyles with ICR beteween bilateral ICR group and unilateral ICR group. But there were some differences in signs and symptoms between patients with bilateral and those with unilateral ICR.I described the difference in symptoms and signs between the two groups in detail according to your suggestion.But it was difficult to describe the morphological changes in this two groups separately. Conclusions has been revised to” Most patients with ICR had signs and symptoms of TMD. The prevalence of clicking and opening-closing deviation were significantly different between the bilateral and the unilateral ICR groups.The size of the condyles decreased significantly,the condyles twisted inward, moved forward, and inclined posteriorly in patients with ICR. “
Introduction: I would suggest you to define ICR according to the TMD-DC.
Response 5:TMD/DC is the international standard for the assessment of TMDs. The TMD/DC is validated for several diagnoses as based on a standardised assessment protocol including history and clinical examination. A diagnostic algorithm utilising both history and clinical data permits to have very high sensitivity and specificity for some TMD subgroups but not for ICR.Radiological examination is important in the diagnosis of ICR. ICR belongs to the classification of joint degenerative diseases in TMD/DC,but there is no more specific definition.
Page 2, lines 52-53: "it's difficult to define" should be omitted.
Response 6:Lines 52-53:Thank you for your advice. "it's difficult to define" has been omitted.
Aim of the study: please specify the "clinical features" in the aim of the study.
Response 7: I specified the "clinical features" in the aim of the study as below: "The aim of this study was to examine the clinical features of ICR such as susceptible population, the signs and symptoms of TMD and the detailed changes of condylar morphology in patients with ICR. We hypothesized that ICR has characteristic clinical and imaging manifestations.”
Methods:
Page 2, line 81: the name of the ethical committee is not reported.
Response 8:The name of the ethical committee has been added to the manuscript.
“All Participants gave their informed consent for inclusion. The study was approved by the Medical Ethics Committee of Stomatology Hospital, Zhejiang University School of Medicine(Approval No. 07). Patients with ICR were recruited from the Department of Orthodontics and Department of Oral and Maxillofacial Surgery of Stomatology Hospital, Zhejiang University from 2014 to 2020. ”
The sample size has been calculated?
Response 9:The sample size hasn’t been calculated before we included all the cases. Because the incidence of ICR is relatively low, the sample size was the number of all the patients that we can collect. However, according to the sample size calculation formula, setting significant level α=0.05, the prevalence of TMD symptoms among Chinese female adults π=30%, maximum permissible error δ is about 11.6% in the calculation of TMD symptoms prevalence in the population with ICR, varying according to different symptoms(clicking, opening-closing deviation, etc.); the maximum permissible error δ is about 25% of estimated standard deviation in the measurements of condylar morphology.
The condyle measurement protocol/cephalometric analysis have been proposed/validated by previous studies?
Response 10: Most of the condyle measurement protocol/cephalometric analysis have been proposed/validated by previous studies. I have improved some measurement methods on the basis of previous research and added references to several unusual measurement methods. Such as the measurments of condylar volume and condylar neck angle.
Results:
Two subheadings should be created: unilateral and bilateral ICR.
Response 11:Your suggestion is very good,according to your suggestion,I have created two subheadings (unilateral ICR group and bilateral ICR group) in results and described the study results separately.
It should be interesting to correlate the radiographic findings (consular height) with clinical signs (open bite, mandibular deviations).
Response 12:This article focused on the study of the TMD symptoms/signs and morphological characteristics of the condyles in patients with ICR,due to the length of the manuscript,the dentofacial characteristics of ICR cases were not involved.Thank you for your advice,we will describe this part in other manuscript.
You assessed three-dimensionally just the volume; the rest of the analysis consisted of 2D measurements. I believe that the condylar morphology should be deeply studies with 3D data.
Response 13:We know that ICR should be deeply studied.After finding a better measurement method,we will try 3D measurments in the further study of ICR.
Discussion:
It should be divided in three sections: interpretation of the results, generalization of the results in the context of the available literature, strengths and limitations of the study.
Response 14: Each paragrgh in the discussion generally includes two sections: interpretation of the results,generalization of the results in the context of the available literature.In the last paragraph of discussion ,I have described the strengths and limitations of the study in detail.
The importance of CBCT is clear and evident.
Response 15: :Yes, I do agree it. I have moved the first paragraph of the discussion(about CBCT importance) and do a brief summary in the introduction section.
CONCLUSION: I do not understand how you demonstrated that the condyles twist inward in ICR.
Response 16:The axial angle in the condyles with ICR was significantly small compared with the control ones.So I mean the condyles rotate inward in ICR.Maybe “rotate inward”is more appropriate to describe this feature?
The study is not reported according to the Strobe guidelines
Response 17:The study is not a typical case control study,so we didn’t report it according to the Strobe guidelines.However,we will try our best to describe the necessaary elements of clinical descriptive study.If there are any deficiencies,please put forward your valuable advice again.
Reviewer 2 Report
Careful editing and proofreading of the complete manuscript are advisable to correct some mistyping. For example: "signs and symptoms," (line2), Background:Little" (line 28), "Methods:Sixty" (line 32), “examined.Signs” (line 34), “described,three-dimensional”(line 34), “analyzed .Results:Symptoms” (line 35), “with ICR.The” (line 37), “controls (P<0.05) .Conclusions:Most” (line 41) and others. The current phrasing makes comprehension difficult. In Table 4 “control group (n=82)” and “Bilateral ICR group(n=82)”, change n=82 by n=41.
Missing information must be supplied. “Medical Ethics Committee of ()” (line 81) and “Department of Orthodontics and Department of Oral and Maxillofacial Surgery of ()”(line 82).
Parts of the manuscript could be reorganized. For example, lines, 302 to 310 should be part of the introduction rather than the discussion section.
Author Response
Reviewer 2
Careful editing and proofreading of the complete manuscript are advisable to correct some mistyping. For example: "signs and symptoms," (line2), Background:Little" (line 28), "Methods:Sixty" (line 32), “examined.Signs” (line 34), “described,three-dimensional”(line 34), “analyzed .Results:Symptoms” (line 35), “with ICR.The” (line 37), “controls (P<0.05) .Conclusions:Most” (line 41) and others. The current phrasing makes comprehension difficult.
Response 1:Thank you for pointing out these problems, it is indeed our negligence.I have corrected these phrasing mistakes in the manuscript and double checked the whole manuscript.
In Table 4 “control group (n=82)” and “Bilateral ICR group(n=82)”, change n=82 by n=41
Response 2:I have measured the left side and the right side of condyles in the control group and Bilateral ICR group. So in these two gruops, we collected the data of 82 condyles, So we specify n=82.
Missing information must be supplied. “Medical Ethics Committee of ()” (line 81) and “Department of Orthodontics and Department of Oral and Maxillofacial Surgery of ()”(line 82).
Response 3:The missing information has been added to the manuscript.
“All Participants gave their informed consent for inclusion. The study was approved by the Medical Ethics Committee of Stomatology Hospital, Zhejiang University School of Medicine(Approval No. 07). Patients with ICR were recruited from the Department of Orthodontics and Department of Oral and Maxillofacial Surgery of Stomatology Hospital, Zhejiang University from 2014 to 2020. ”
Parts of the manuscript could be reorganized. For example, lines, 302 to 310 should be part of the introduction rather than the discussion section.
Response 4:Thank you for your valuable suggestions.I have moved the first paragraph of the discussion(about CBCT importance) to the introduction section.
Reviewer 3 Report
This study addressed the the signs and symptoms analysis of temporomandibular dysfunction in patients with idiopathic condylar resorption associating with the morphological and positional characteristics of the condyles.
However, I found some concerns in this manuscript as follows:
Introduction:
The first thing I would encourage the authors to emphasize is that despite CBCT and CT may be recommended in some cases for TMJ evaluation, the MRI remains as the gold standard examination for TMJ features analysis due to its high-contrast sensitivity to tissue differences and the absence of ionizing radiation (https://www.ncbi.nlm.nih.gov/pmc/articles/PMC3513803/), and should be preferably recommended whenever possible. This statement is important to increase the quality of the manuscript regarding ethical standards.
Materials and Methods
Participants:
Signs and symptoms of ICR were obtained through clinical consultation and examination performed by an experienced doctor: TMD specialist? There is a description of the clinical examination, but some reference is missing.
Image measurements:
Examiner: Radiologist? Years of practice? Where were the analyses performed (computer details)?
Discussion:
The Discussion is slightly disorganized, and it is difficult to determine what points are based on the author's study, their conjecture, or previously published literature. The
Limitations section should be addded to include concerns raised in Weaknesses.
Author Response
Reviewer 3
This study addressed the the signs and symptoms analysis of temporomandibular dysfunction in patients with idiopathic condylar resorption associating with the morphological and positional characteristics of the condyles.
However, I found some concerns in this manuscript as follows:
Introduction:
The first thing I would encourage the authors to emphasize is that despite CBCT and CT may be recommended in some cases for TMJ evaluation, the MRI remains as the gold standard examination for TMJ features analysis due to its high-contrast sensitivity to tissue differences and the absence of ionizing radiation (https://www.ncbi.nlm.nih.gov/pmc/articles/PMC3513803/), and should be preferably recommended whenever possible. This statement is importanht to increase the quality of the manuscript regarding ethical standards.
Response 1:Thank you for your valuable advice.MRI actually is the gold standard examination for TMJ features analysis due to its high-contrast sensitivity.It has obvious advantages in observing the soft tissue around the TMD,especially the condyle-disc relationship.We have a further research plan: study the changes of condyle-disc relationship in ICR patients by MRI.But this manuscript mainly concern the bony characteristics of condyle in patients with ICR,so we use the CBCT data here.
Materials and Methods
Participants:
Signs and symptoms of ICR were obtained through clinical consultation and examination performed by an experienced doctor: TMD specialist? There is a description of the clinical examination, but some reference is missing.
Response 2:The experienced doctor performed TMJ examination is an orthodontist with TMD professional background.She is a member of the TMJ professional commitee of Chinese Stomatological Association and is a doctor of the joint clinic of TMD and orthodontics.
I have add a reference about the clinical examination of TMD.
Image measurements:
Examiner: Radiologist? Years of practice? Where were the analyses performed (computer details)?
Response 3:We use Dolphin Imaging program to do the measurement. Dolphin Imaging program is a professional orthodontics software.So the examiner is not a radiologist but a orthodontist (more than 5 years practice) who can use the Dophin software skillfully.
The computer details: LAPTOP-MK4VN6T6 Intel(R) Core(TM) i5-10210U CPU @ 1.60GHz 2.11 GHz
Discussion:
The Discussion is slightly disorganized, and it is difficult to determine what points are based on the author's study, their conjecture, or previously published literature. The Limitations section should be addded to include concerns raised in Weaknesses.
Response 4:I have reorganized the discussion section.Each paragragh in the discussion generally includes two sections: interpretation of the results,generalization of the results in the context of the available literature.The last paragraph in discussion ,I have described the strengths and limitations of the study in detail.
Round 2
Reviewer 2 Report
The authors have satisfactorily responded to all my questions and made the necessary changes to the manuscript
Author Response
Thanks for your kindly reply!
Reviewer 3 Report
I have reviewed the re-submission and the authors have carefully amended their manuscript following the additional reviewers suggestions.
Author Response
Thanks for your kindly reply!